# Robot Learning from Demonstration: Enhancing Plan Execution with Failure Detection Model

## Abstract

Learning plans from demonstrations has emerged as a valuable paradigm, in which a robot autonomously completes a task by executing a sequence of actions according to a learned plan. Nevertheless, the execution of an action may encounter failures in the real environment, such as failing to pick up a cup, resulting in plan execution failure. The execution of a broken plan may damage the environment, e.g., cooking coffee when a cup is not successfully placed. To avoid such risks, action failure detection is crucial. However, the action failure within the execution of task plans is often neglected in existing research. To address the problem, we propose a framework that learns an executable plan that checks failures of each action, called failure-aware plan. Our framework employs meta-learning to learn neural network-based failure-aware task plans. Initially, by using trajectory data collected from robot randomness execution, the framework pre-trains a model that discriminatively captures the state features of various actions at different stages. Utilizing user demonstration trajectories labeled as either success or failure, the pre-trained model undergoes fine-tuning, which is then employed to determine the success or failure of an action execution by means of the corresponding state features. We demonstrate the effectiveness of our approach through experiments on a robot in a simulation environment. Our approach outperforms the compared method when only limited demonstration data is available. This work contributes to enhancing the reliability of plan execution for robot by considering action failure detection.

## 1 Introduction

The use of robots is increasingly prevalent in both everyday life and industrial production (Havoutis & Calinon, 2019; Hung & Yoshimi, 2017). Learning plans from demonstrations in robotic field is a paradigm for enabling robots to autonomously perform tasks. For example, consider the common task of 'making coffee', which comprises a series of actions: picking up a cup, positioning it under the spout, pressing a button, and subsequently retrieving the cup. After formulating a plan, the subsequent challenge lies in its execution. As actions are carried out in real-world settings in accordance with a plan, they may encounter failures during plan execution, as highlighted in Karapinar et al. (2012). The failures may arise due to the deviations from expected outcomes of robot actions, caused by real-world noise. More specifically, when a grasp action is scheduled to executed during the task of 'making coffee', unforeseen scenarios like the object slipping and falling to the ground may occur. If the plan proceeds with execution, the result may be the spillage of hot coffee, potentially resulting in a significant safety hazard. Thus, the failure of robot action execution within a plan can potentially have significant impacts on the environment and may even pose safety risks. While research exists that focuses on learning plans from demonstrations, such as symbolic representations learned for constructing plans (Konidaris et al., 2018; Ames et al., 2018), various robotic application for task-level planning (Ekvall & Kragic, 2008; Hayes & Scassellati, 2016; Yin et al., 2019). They generally overlook the consideration of action failure detection during plan execution.

However, this is a challenging task that determines the effect of scheduling action execution status based on the current environment in which the robot performs actions. Given that the robot's operational environment is a continuous, high-dimensional space encompasses both the robot's intro-

spective states (e.g., force-torque, velocity, and tactile) and environmental states. Existing research focuses on the assessing the execution of a standalone action instead of a whole plan (Pettersson, 2005), e.g., gearbox failure detection for industrial robots (Vallachira et al., 2019), manipulation failure detection for robots (Inceoglu et al., 2021; 2023). An inherent challenge lies in the fact that the effect of each action in a plan execution can vary based on the contextual environment. A solely learned action failure detector may not achieve successful generalization to the plan. Implementing failure detection for every action in a plan necessitates the creation of datasets for each task-specific action, incurring substantial costs. This challenge stems from concerns about the sample efficiency of robotic manipulation and the reliance on multimodal data from specific sensors or devices.

To address the challenge, we propose a framework which employs meta-learning to learn an executable plan that checks failure of each action by learning from few demonstrations, called failure-aware plan. Our approach follows a two-step methodology to construct executable action sequences from user demonstrations as depicted in Figure 1. Initially, our framework learns a *Action Model* and a pre-trained *Detect Model*, using trajectory data collecting by robot randomness execution. In the first step, trajectories obtained from user demonstrations are input to a sequence learning mechanism with *Action Model*.T This mechanism is utilized to segment user demonstrations into individual action fragments. In the second step, the *Detect Model* is fine-tuned through the corresponding action fragments. The fine-tuned model are able to analyze the state features during execution of actions and is responsible for failure detection for the corresponding action. Simultaneously, action controllers are learned from the segmented action fragments for robots to perform a robotic manipulation. By taking the above steps, we can derive a failure-aware plan in which each action execution incorporates model-based failure detection.

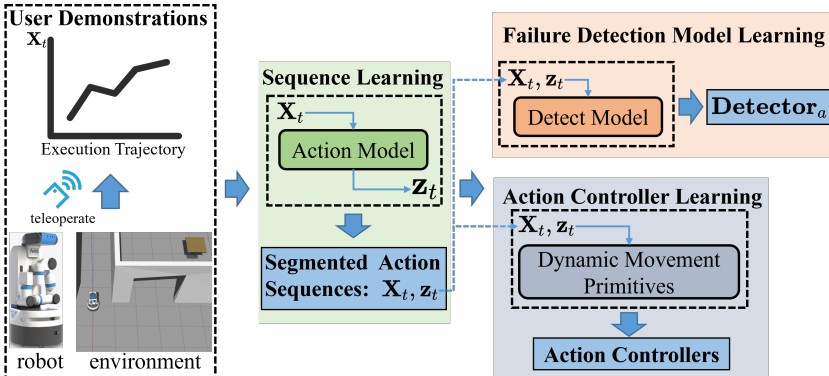

Figure 1: The overview of our methodology. The end user teleoperates the robot to perform a task and the demonstration trajectories are recorded. The demonstrations are segmented into action fragments by *Action Model*. *Detect Model* is fine-tuned by meta-learning. Simultaneously, the action implementations are learned by Dynamic Movement Primitives.

We implement our approach on a 'Fetch' robot for a pick-and-place demonstration in a 'Gazebo' based simulation environment. Subsequently, we conducted experiments under varying settings to validate its performance. The case study demonstrates that our approach effectively learns failure-aware plans, enabling accurate evaluation of robot action execution within task environments. Moreover, our experiments conducted across various settings highlight our approach's ability to obtain failure detection of high accuracy. Notably, our approach outperforms an baseline approach, especially when only a limited number of demonstrations are accessible.

## 2 RELATED WORK

Learning plans from demonstrations in the robotic field can be summarized as task-level learning and planning for robotic application (Ekvall & Kragic, 2008; Hayes & Scassellati, 2016; Yin et al., 2019). Complex task plans contain several subtasks that exhibit specific ordering constraints and interdependence, which are learned by incremental learning method (Grollman & Jenkins, 2010), method for learning sequential robot skills through kinesthetic teaching (Manschitz et al., 2014),

online algorithm for constructing skill trees (Konidaris et al., 2012; Lioutikov et al., 2018), interaction paradigm (Mohseni-Kabir et al., 2019). Nevertheless, the current body of literature neglects the aspect of identifying action failures that may occur while a plan is being executed.

The failure detection for a standalone action execution of the robot has been studied (Pettersson, 2005; Chalapathy & Chawla). Kalman Filter (Goel et al., 2000), residual based systems Stavrou et al. (2016) Non-parametric Bayesian models (Zhou et al., 2020) are used for action execution monitoring in robotics. A multimodal execution monitoring system is proposed for the assistive feeding task in Park et al. (2018). In Vallachira et al. (2019), introspective data are used to detect gearbox failures for industrial robots. Multimodal data is also considered in Inceoglu et al. (2021; 2023). Each of the methods relies on a dataset that is meticulously customized to meet its specific requirements, for which the constructing is costly. Furthermore, the dataset is employed to learn failure detection model for a standalone action. The context of the environment, where the plan executes, should be considered when learning the action failure detector. In our approach, we utilize meta-learning to learn task-relevant failure detection of the action from few user demonstrations, which improves the efficiency.

## 3 APPROACH

Our approach is illustrated in Figure 1. The approach comprises three main components: sequence learning, action controller learning, and failure detection model learning. These components collectively enable the learning of a failure-aware plan from few user demonstrations. The sequence learning module segments a small number of user demonstration trajectories into action fragments. These trajectories of the action fragments serve a dual purpose. First, they are used to learn the controllers for the actions in the plan. An executable plan can be synthesized by combining the segmented action sequences with their respective controllers. Second, we utilize the segmented data for fine-tuning, employing meta-learning algorithms to obtain task-relevant action failure detection models.

### 3.1 FORMALIZATION

At a high level, a plan with action sequences can be defined as $\Pi = \langle a_1, a_2, \ldots, a_K \rangle$, where $K$ is the length of the plan. We define an operator $Detector_a(s) \to \{success, failure\}$ as a failure detector for action $a$, e.g., it returns $success$ if the state $s$ reflects the correct execution of the action $a$. Each action also has an operator $Controller_a$ to perform a robotic manipulation. From a low-level perspective, consider $\{\mathbf{X}_t\}_{t=1:T_m}$ as a robot execution trajectory where $\mathbf{X}_t \in \mathbb{R}^D$ represents a state of the environment (including the robots) at time point $t$ and $T$ is the length of the trajectory. Each state $\mathbf{X}_t$ is sampled from low-level sensor readings. We assume that the observations of the execution trajectory of an action reflect the impact on the robot itself and the environment. We denote the *Detect Model* $f_D(\{\mathbf{X}_t\}_{t=i:j}) \to \{1, 0\}$ to model the $Detector_a(s)$ in low level. An execution trajectory is obtained through the execution of a composite set of fundamental robotic controllers, such as mobile platform movement, rotation, and manipulation of joint angles in the robot arm. We assume access to a supervisor that assigns complex action labels to the trajectories, resulting in a set of labeled trajectories $D_M = \{\mathbf{X}_{m,t}, \mathbf{z}_{m,t}\}_{m=1:M,t=1:T_m}$. The action label $\mathbf{z}_{m,t} \in \{1 \ldots C\}$ describes a specific task such as picking or grasping where $C$ is the number of action labels. Without loss of generality, we drop the indices $m$ to denote a state as $\mathbf{X}_t$ and the action label $\mathbf{z}_t$ for the rest of the paper.

An execution trajectory for a particular task is defined as $Traj = \{\mathbf{X}_t\}_{t=1:T}$. First, we seek to learn an action model $f_A : \mathbf{X}_t \to \mathbf{z}_t$ based on $D_M$ to predict the action label $\mathbf{z}_t$ given a subsequence of the trajectory $\{\mathbf{X}_t\}_{t=i:j}$, where $\mathbf{X}_t \in \mathbb{R}^{D \times (j-i+1)}$ and $\mathbf{z}_t \in \{1 \ldots C\}$. In this way, a given set of trajectories for a particular task can be segmented into a set of subsequences $D_N = \{\mathbf{X}_{n,t}, \mathbf{z}_{n,t}\}_{n=1:N,t=1:T_n}$ with $N \ll M$. Through a meta-learning approach, the *Detect Model* $f_D$ is pre-trained by utilizing the dataset $D_M$, subsequently fine-tuned based on the smaller dataset $D_N$.

## 3.2 Sequence Learning

We capture the spatio-temporal dependency in the execution trajectories to predict action segments using a sequence learning approach. We use the *Action Model* to model the mapping from state observations to action labels with the probability $P(\hat{\mathbf{z}}_{t:t+l} \mid \mathbf{X}_{t:t+l})$ using a stride of length $l$ within a mini-batch. The architecture of the *Action Model* is a recurrent neural network (RNN) which maintains an additional hidden state and uses the previous hidden state and the current input $\mathbf{X}_t$ to produce a new hidden state and the output $\hat{\mathbf{z}}_t$. The hidden state preserves the effect of the previous observation in predicting the current output. We use bidirectional Long Short-Term Memory (LSTM) in *Action Model*, which also preserves the effect of future observations within a sequence. We minimize the cross-entropy loss between the predicted action label $\hat{\mathbf{z}}_t$ and the ground truth. We infer the most likely action fragment $\hat{\mathbf{z}}_{i:j}$ by maximizing the probability $P(\hat{\mathbf{z}}_{i:j} \mid \mathbf{X}_{i:j})$ through sliding a window of width $l$ along a trajectory. The prediction for each observation fragment is transformed into prediction for each time point $i$ as follows:

$$\mathbf{z}_i = \arg\max_{\mathbf{z}} \sum_{t=i-l}^{i} \delta(f_A(\mathbf{X}_{t:t+l}), \mathbf{z}_{\{1...C\}}),$$

where $\delta$ is a function used to count the predicted value of each action category. Through the method of sequence learning, the sequence labels $\hat{\mathbf{z}}_{1:T}$ of an unlabeled trajectory are estimated. The compilation of labeled fragments $D_N$ is accomplished by linking the observations from each time point.

## 3.3 Action Controller Learning

We use user demonstrations to learn action controllers to reproducing the robot motion planning. We apply Dynamic Movement Primitives (DMPs) to encode a movement trajectory in terms of a dynamics of nonlinear differential equations. Only a small number of parameters are required to model the demonstration trajectory, through which the operations can be quickly reproduced. DMPs further facilitate the generalization and modification of the original trajectory by introducing additional task parameters during the reproduction of the demonstration trajectory, such as changing the amplitude and frequency of the joint angle curve, changing the starting and target positions of the end trajectory of the manipulator, etc., The DMPs equations (Park et al., 2008) are defined as follows :

$$\tau^2 \ddot{y} = \alpha_y(\beta_y(g - y) - \tau\dot{y}) + Kf - Kx(g - y_0),$$

in which the nonlinear item $f$ is defined as:

$$f(x) = \frac{\sum_{i=1}^{N} \Psi_i(x)\omega_i x}{\sum_{i=1}^{N} \Psi_i(x)},$$

where the $\Psi_i$ is Gaussian function and $\omega_i$ is the corresponding weight. The parameters $\alpha_y$, $\beta_y$, $K$, $g$, $y_0$ are used to adjust the shape of the trajectory. The canonical system is defined as $\tau\dot{z} = -\alpha_z z$ with $\alpha_z$ is a constant. Based on the $D_N$, we learn the parameters of DMPs for each action label respectively. Given $y_{demo}, \dot{y}_{demo}, \ddot{y}_{demo}$, we can obtain the nonlinear function that needs to be fitted as:

$$f_{target} = \frac{\tau^2 \ddot{y}_{demo} - \alpha_y(\beta_y(g - y_{demo}) - \tau\dot{y}_{demo})}{K} + x(g - y_0)$$

We construct the loss function as follows:

$$J_i = \sum_{t=1}^{P} \Psi_i(t)(f_{target}(t) - \omega_i \xi(t))^2$$

In this way, DMPs for actions assignments can be learned from $D_N$. They can be formulated as APIs in more flexible manipulation task.

## 3.4 Failure Detection Model Learning

We employ the *Detect Model* to check the execution of a specific action by analyzing the corresponding observations. In this study, end users only perform a small amount of demonstrations. After conducting sequence learning on the trajectories, we obtain a small dataset denoted as $D_N$,

which might be insufficient for training the *Detect Model*. Therefore, we initially pre-train the *Detect Model* utilizing the more extensive dataset $D_M$ in a meta-learning manner. Subsequently, we fine-tune $f_D$ using the smaller dataset $D_N$. To capture the features of observations, we employ a transformer encoder in *Detect Model*. Transformers have achieved superior performances in many tasks in the time series data such as forecasting and classification. They possess the capability to learn long-range dependencies present within execution trajectories. By employing this architecture, we create specific *Detect Models* as failure detectors for individual actions.

Essentially, the *Detect Model* functions as a binary classifier, responsible for determining the success or failure of a given action execution. The dataset $D_M$ is processed to facilitate the sampling of multiple binary classification tasks for pre-training. The same procedure is followed for the smaller dataset during the fine-tuning phase. Each sample within $D_M$ is divided into three distinct groups. For example, in the case of the **grasp** action, the observations are categorized into the start stage, middle stage, and end stage. These stages correspond to the initiation of the grasping action, the process of moving the robot arm, and the final state of successfully grasping the object, respectively. This enables the *Detect Model* to capture action-specific characteristic from various stages of state observations. Consequently, we obtain the dataset $\{\mathbf{X}_{k,t}\}_{t=1:T_k, k=1:C \times 3}$. During the pre-training phase, two categories are randomly chosen from the total of $C \times 3$ categories for each learning task. Subsequently, these two categories are assigned binary labels to facilitate the pre-training of the *Detect Model*. The model weights are updated using second-order differentiation to enhance the sensitivity of the loss function towards the newly constructed tasks (Finn et al., 2017). In the fine-tuning stage, the same idea is applied to process the dataset $D_N$ into a fine-tuning set. This processed data can be utilized to fine-tune the *Detect Models* associated with corresponding actions for failure detection. For instance, consider the case of learning a failure detector for a *grasp* action. In this scenario, the positive samples within the fine-tuning dataset will include observations from the final stage of the *grasp* action, where the successful execution of the action is expected. On the other hand, the remaining data, which encompasses other stages of the *grasp* action or unrelated actions, will serve as negative samples. This fine-tuning process enhances the *Detect Model*'s ability to accurately identify success or failure in specific action during task executions, catering to the unique characteristic of each action type. Ultimately, $Detector_{grasp}(s)$ is obtained for detecting failures in the execution.

## 4 EXPERIMENTS

### 4.1 ENVIRONMENT SETTINGS

We implement our approach on a 'Fetch' robot within the "Gazebo" simulation environment (Koenig & Howard, 2004). The 'Fetch' robot is a commonly employed research platform for validating techniques in the field of robotics, encompassing a wide range of applications (Liu, 2020; Chen et al., 2020). A 'Fetch' robot is mainly equipped with a mobile base and a robotic arm. The mobile base comprises two hub motors and four casters, while the arm features a seven-degree-of-freedom design along with a gripper. Users are provided with a set of fundamental APIs to control the robot's movements, including forward and backward motion, rotation of the mobile base, adjustment of the arm's joint angles, and manipulation of the gripper's opening and closing. 'Gazebo' provides the capability to accurately and efficiently simulate groups of robots within intricate indoor and outdoor environments. We implement our approach within a scenario created in the 'Gazebo' environment, which is known for its high-fidelity simulation capabilities. The execution of actions within the simulation environment introduces uncertainty stemming from factors such as the physics engine or probability-based algorithms. For instance, an object may accidentally slip from the robot's gripper. Given the high-fidelity nature of the simulator, we maintain confidence that our approach can be extended to real-world environments featuring physical hardware, even after testing within the 'Gazebo' simulation environment.

We have designed a pick-and-place task scenario comprising four steps. Consider a simulation environment containing two tables: *TableA* and *TableB*, depicted in Figure 2. *TableA* is positioned outside a room, and it features a cube placed atop its surface. On the other hand, *TableB* is located within the room. The 'Fetch' robot is initially directed to approach *TableA*, after which it is instructed to pick up the cube. Subsequently, with the cube in its possession, the robot is guided to transition to *TableB*, where it completes the task by placing the cube on the table. An end user is

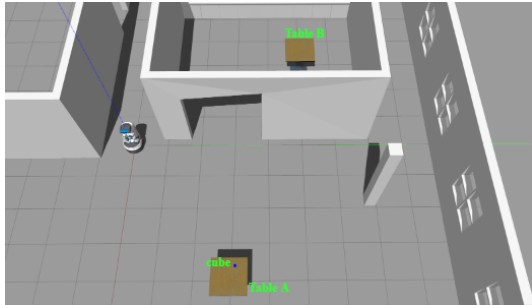

Figure 2: The scenario for pick-and-place task in 'Gazebo'. A start point is the initial position of the robot. The *TableA* with a cube on it is placed outside. The *TableB* is placed inside the room.

equipped with a remote controller for the 'Fetch' robot, allowing them to guide the robot through the task by controlling its movement, rotation, and arm joint angles.

We provide a concrete example to illustrate how our approach learns executable action sequences with failure detection by learning from a small number of demonstrations. A detailed description is available in the supplementary material.

## 4.2 EXPERIMENT SETTINGS

We conduct experiments with multiple different settings to answer the following research questions.

- **RQ1.** How accurate is our approach when applied to action failure detection?

- **RQ2.** what is the impact of the number of demonstrations on the accuracy of our approach?

- **RQ3.** How generalized is our approach when applied in changed scenarios?

In this context, the *Detect Model* is learned from demonstrations conducted during the experiments. The evaluation of the action failure detection are categorized into four classifications:

- *TP (true positive)* denotes a positive evaluation (indicating truth) for an action that is executed correctly within 'Gazebo'.

- *TN (true negative)* denotes a positive evaluation (indicating falsehood) for an action that is executed abnormally within 'Gazebo'. For instance, if the 'Fetch' robot drops the cube while attempting to pick or transport it due to physics engine randomness, the action failure detection evaluation yields a false value.

- *FP (false positive)* denotes a negative evaluation (indicating falsehood) for an action that is successfully completed.

- *FN (false negative)* stands for a negative evaluation (indicating truth) for an action that is executed abnormally.

A learned plan is executed for a request that has the same goal as the demonstrations. An execution is considered a success if the action sequence is executed as expected, with all action failure detections evaluated to be positive, i.e., all *TP*. However, an execution failure may be caused by the detection of an abnormal action (*TN*), an incorrect evaluation of an abnormal action (*FN*), or an incorrect evaluation of a completed action (*FP*). We consider that *TN* is desirable in task plan execution because the execution can be terminated as soon as an action abnormality is detected. *FN* and *FP*, on the other hand, cause errors in dispatching actions at runtime, i.e., executing a subsequent action that should not be executed, or failing to execute a subsequent action that should be executed.

We count the number of the correctly evaluated failure to compute the accuracy. Suppose *cnt* is the total count of action failure detection evaluation performed during the execution in an experiment, the accuracy is calculated by (*#TP+#TN*)/*cnt*, where *#* means the count.

## 4.3 RQ1: ACCURACY OF FAILURE DETECTION

This experiment is used to show the correctness of the action failure detections of the learned plan towards different task requests. We design two types of requests according to different environment settings. The variable points in the settings are the changes in the size and position of table(s) and cube(s) while the room layout remains the same. One setting is to require the robot to perform a task repeatedly in the same environment as the demonstrations. We name it as *stationary-environmental request (SeR)* in which the positions of the tables and the cube on the table are fixed. The other setting is to require the robot to perform a task repeatedly but in changed environment, i.e., the positions of the tables and the cube are different for each execution. We name it as *variable-environmental request (VeR)*. We apply our approach with 5 demonstrations both in *SeR* and *VeR*.

Additionally, we carry out an experiment where we omit the fine-tuning process and instead directly employ the pre-trained *Detect Model* as a general action failure detector. This serves to substantiate the necessity of conducting individualized fine-tuning for each action. We consider the approach from Konidaris et al. (2018) to compare with our approach. The referenced study focuses on capturing the planning domain through large number of demonstrations, wherein the precondition and postcondition are acquired using a rule-based algorithm. We employ the similar network as our method to learn action failure detectors by the descriptions of the learned plan. The comparison aims to demonstrate the effectiveness of our method, which employs a meta-learning algorithm and learns from only a few demonstrations. More details of the implementation are detailed in the supplementary material. To compare with the model-based action failure detection of our approach, we evaluate the baseline approach using both 5 and 100 demonstration instances for evaluations separately.

Table 1 lists the accuracy for evaluating the action failure detection using our approach in typical demonstration count settings. The 'baseline (5)' and 'baseline (100)' represent the case of the baseline approach trained by 5 demonstrations and 100 demonstrations respectively. The symbol 'Ours (no-ft)' represent our method without the fine-tuning phase for *Detect Model*. Each of the approaches is executed 50 times for *SeR* and *VeR* respectively. *#ES* represents the count of the success in the 50 executions. *#EF* represents the count of the failure caused by a *TN* in the 50 executions.

Our approach achieves high accuracy for both *SeR* and *VeR*. In particular, we consider the failure detectors learned by our approach has adaptability in a unexperienced environment. The accuracy for *VeR* is slightly lower than that for *SeR*. This is because the probability of the incorrect evaluations increases in changed environment settings. Furthermore, 28 out the 50 requests are completed successfully in *SeR* while the number declines to 19 in *VeR*. The decline is reasonable in changed environment since the probability that all the learned action failure detectors involved are evaluated correctly becomes smaller.

The accuracy of the rule-based approach for *SeR* increases with the counts of demonstrations. In the case of 5 demonstrations, the accuracy is lower than our approach by about 10%. However, the number of the completed requests (*#ES*=2) is significantly lower. It means that the rule-based approach under 5 demonstrations is hardly to deal with the user's requests. In addition, the accuracy and *#ES* are comparable with those by our approach when the rule-based approach is applied based on 100 demonstrations. It indicates that our approach outperforms the rule-based baseline approach when only few demonstrations are available.

On the other side, the accuracy of the rule-based approach is quite low for *VeR*. The reason lies in that the rule-based approach learns the state characteristic of actions especially related to the demonstration trajectories. The characteristic is not generalized for the learned detector when evaluated for the new requests in different environment settings.

Compared with our approach fine-tuned by 5 demonstrations, our method without fine-tuning phase (Ours (no-ft)) achieves a low accuracy either in *SeR* or *VeR*. Besides, in this case, the value of *#ES* is zero both in *SeR* and *VeR*. This is because the pre-trained *Detect Model* is only trained in task-agnostic offline phase so that it is hard to deal with task related features. This explains why our approach needs fine-tuning to learn task-related action failure detectors.

Table 1: Accuracy of evaluating action failure detection of different approaches

|  |  | cnt | #TP | #TN | #FP | #FN | Accuracy | #ES | #EF |
|---|---|---|---|---|---|---|---|---|---|
| baseline | *SeR* | 137 | 95 | 12 | 30 | 0 | 78.0% | 2 | 14 |
| (5) | *VeR* | 79 | 29 | 3 | 47 | 0 | 40.5% | 0 | 3 |
| baseline | *SeR* | 182 | 153 | 12 | 10 | 7 | 90.7% | 21 | 12 |
| (100) | *VeR* | 78 | 32 | 6 | 40 | 0 | 48.7% | 4 | 6 |
| Ours | *SeR* | 121 | 71 | 5 | 42 | 3 | 62.8% | 0 | 5 |
| (no-ft) | *VeR* | 80 | 30 | 8 | 39 | 3 | 47.5% | 0 | 8 |
| Ours (5) | *SeR* | 152 | 130 | 8 | 8 | 6 | 90.8% | 28 | 8 |
|  | *VeR* | 151 | 120 | 11 | 15 | 5 | 86.8% | 19 | 11 |

## 4.4 RQ2: Impact of the Number of Demonstrations

This experiment evaluates the impact of the number of demonstrations on our approach from two levels. First, the convergence of fine-tuning of the *Detect Model* is evaluated under different demonstration counts. Second, the accuracy of action failure detection is further evaluated based on different numbers of demonstrations. To this end, we learn from 1, 5, 10 and 20 demonstrations. We expect to reveal the convergence performance of a *Detect Model* in each updating step by evaluating the accuracy of the test set under different demonstration counts.

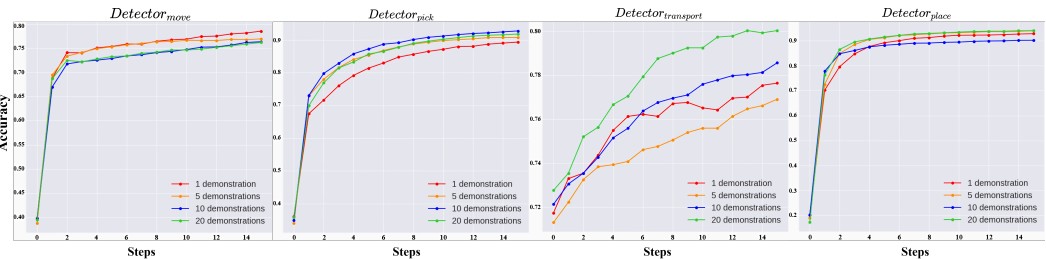

Figure 3: The convergence of fine-tuning *Detect Model* under different demonstration counts with 10 random seeds.

Figure 3 visualizes the convergence of the fine-tuning phase of *Detect Model*. As the number of the updating steps increases, most of the *Detect Models* can converge quickly. The $Detector_{move}$, the $Detector_{pick}$ and the $Detector_{place}$ get converged by about 10 gradient descent steps, while the $Detector_{transport}$ almost gets converged by 15 steps. It validates the effectiveness of the approach conducting meta-learning.

The accuracy of the $Detector_{pick}$ and the $Detector_{place}$ reaches approximately 90%. However, the accuracy of the $Detector_{move}$ and the $Detector_{transport}$ is relatively low. We find that the failure detection of the actions performed by the robotic arm (*pick*, *place*) are evaluated better compared with those performed by the robot's mobile base (*move*, *transport*). We consider the difference comes from the different number of state features related to an action. The more state features associated with an action, the better state characteristic of the action can be learned. For example, the action *pick* involves more state features than the action *move* since the robotic arm provides more features than the mobile base.

In addition, we find that the pre-trained *Detect Model* fine-tuned by more data can achieve a better accuracy in general. However, the accuracy of $Detector_{move}$ behaves differently. The $Detector_{move}$ fine-tuned by one demonstration and five demonstrations outperforms the network fine-tuned by ten or twenty demonstrations. The reason is that the small number of state features associated with the action *move* makes $Detector_{move}$ easier to be trained. In this case, the small amount of fine-tuning data makes $Detector_{move}$ train faster with few gradient descent steps.

For the second evaluation objective, the failure detectors learned through different counts of demonstrations are executed 50 times respectively for the task in the stationary environment. Table 2 lists the accuracy for evaluating results. Overall the accuracy is ascending by the increasing number of

demonstrations. The counts of *TP* gradually increases while the count of *FP* gradually decreases. It indicates the ability of identifying the correctly completed actions is becoming stronger. Therefore, the number of the successfully completed requests **#ES** reasonably increases. Furthermore, the sum of **#FP** and **#FN** decreases roughly. It indirectly indicates the ability of correctly detecting the action abnormalities gets stronger with the increase of the demonstration counts. In the case of learning from one demonstration, 40 out of the 50 executions fail. Among the 40 executions, action abnormalities in 13 executions are correctly detected, i.e., the probability is 32.5%. Comparatively, in the case of learning from 20 demonstrations, the correct verdict towards the abnormalities is 15 out of 23 failed executions. The probability increases to 65%.

Table 2: Accuracy of evaluating action failure detection for *SeR* under different demonstration counts

| #demonstration | $cnt$ | #TP | #TN | #FP | #FN | Accuracy | #ES | #EF |
|---|---|---|---|---|---|---|---|---|
| 1 | 140 | 100 | 13 | 19 | 8 | 80.7% | 10 | 13 |
| 5 | 152 | 130 | 8 | 8 | 6 | 90.8% | 28 | 8 |
| 10 | 169 | 148 | 11 | 10 | 0 | 94.1% | 29 | 11 |
| 20 | 159 | 136 | 15 | 4 | 4 | 95.0% | 27 | 15 |

### 4.5 RQ3: GENERALIZATION IN EXTENDED SCENARIOS

This experiment aims to evaluate the generalization of our approach when reusing action failure detectors learned from a basic scenario in a more complex scenario. To this end, we design a more complicated task which is regarded as an extension of the four-step pick-and-place task. Based on the same scenario, we set up two cubes on *TableA*. We instruct the robot to pick up one of the cube and transport it to *TableB*. Then the robot needs to move back to *TableA* again and pick up another cube and transport is to *TableB*. We retain the detectors learned in the four-step task by 5 demonstrations. Consequently, they are generalized, i.e., $Detector_{move_1}$ for action $move_1$ and $move_2$, $Detector_{pick_1}$ for action $pick_1$ and $pick_2$. Then, an eight-step task $\Pi$ can be assembled as follows: $\langle move_1, pick_1, transport_1, place_1, move_2, pick_2, transport_2, place_2 \rangle$ With this setting, the model-based detectors are further used in the eight-step task for evaluating action failure, which evaluates the generalization of our method. The task is executed for 50 times in *SeR* setting.

Table 3: Accuracy of evaluating action failure detection generalized in the eight-step task

| #demonstration | $cnt$ | #TP | #TN | #FP | #FN | Accuracy | #ES | #EF |
|---|---|---|---|---|---|---|---|---|
| 5 | 259 | 213 | 15 | 14 | 17 | 88.0% | 4 | 15 |

The accuracy for evaluating action failure detection is listed in Table 3. It achieves an accuracy rate of 88.0% which is almost the same as the case learning from five demonstrations in RQ2. Even if the trajectory in the eight-step task is equivalent to the trajectory in the four-step task looping twice, each corresponding segment of trajectory is still somewhat different. Since the *Detect Model* is not learned and trained in completely identical demonstrations. This may lead to the slightly drop of accuracy rate. We also find that the total accuracy rate is not much different, however, the number of **#ES** is reduced a lot in 50 executions. This is because the eight-step task can be considered as a long-horizon task for robotic operations. Since the robot is easily affected by the real environment when performing tasks, there will be a certain probability of failure. However, based on the accuracy rate, when transferring from basic scenario to extended scenario, our approach performs well to show some generalization.

## 5 CONCLUSION

In this paper, we propose a framework for learning neural network-based failure detection for robot action execution. Our approach learns the sequenced actions and their corresponding failure detectors, formulating a failure-aware plan. As a result, this mitigates the occurrence of substantial safety issues during plan execution. The incorporation of meta-learning enhances learning efficiency within a user environment. We conducted various experiments which demonstrates that the reliability of plan execution for robot is enhanced by considering action failure detection.

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
