# SUPPLEMENTARY MATERIAL

## 1 DETAILS FOR ROBOT AND SCENARIO

We showcase the practical implementation of our proposed approach within the dynamic interplay of the 'Fetch' robot and the 'Gazebo9' simulation environment. The 'Fetch' robot is a mobile manipulator designed for research and development purposes in robotics and automation. It's known for its versatility and capability to perform tasks in various environments. The 'Fetch' robot is mainly equipped with a mobile base and a robotic arm. The mobile base is made of two hub motors and four casters, while the arm is a seven-degree-of-freedom arm with a gripper. 'Gazebo9' is a widely-used open-source simulation tool designed for robotics and automation applications. It provides a platform for creating, simulating, and testing robotic systems and environments. 'Gazebo9' enables researchers, developers, and engineers to validate their algorithms and control strategies in a virtual environment before deploying them on real robots.

The key features can be obtained by subscribing to the corresponding ROS topics, which is provided by 'Gazebo9'. In this case, we achieve the state features associated with the robot features and the environment features with the topic *joint_states* and */gazebo/model_states*. The topic *joint_states* provides 13 joints of the robot features including the arm and the mobile base, as shown in the following:

$$l\_wheel\_joint, r\_wheel\_joint,$$
$$torso\_lift\_joint, bellows\_joint,$$
$$shoulder\_pan\_joint, shoulder\_lift\_joint,$$
$$upperarm\_roll\_joint, elbow\_flex\_joint,$$
$$forearm\_roll\_joint, wrist\_flex\_joint,$$
$$wrist\_roll\_joint, l\_gripper\_finger\_joint,$$
$$r\_gripper\_finger\_joint.$$

Each joint incorporates 3 properties including position, velocity and effort. There are $3 \times 13 = 39$ joint features from 'Fetch' body. The topic */gazebo/model_states* provides 13 features associated with the position and the orientation of the robot as well as the property of the cube, listed as following:

$$fetch\_pose\_position\_x, fetch\_pose\_position\_y,$$
$$fetch\_pose\_position\_z, fetch\_pose\_orientation\_x,$$
$$fetch\_pose\_orientation\_y, fetch\_pose\_orientation\_z,$$
$$fetch\_pose\_orientation\_w, fetch\_twist\_linear\_x,$$
$$fetch\_twist\_linear\_y, fetch\_twist\_linear\_z,$$
$$fetch\_twist\_angular\_x, fetch\_twist\_angular\_y,$$
$$fetch\_twist\_angular\_z.$$

There are $13 + 13 = 26$ joint features from the 'Gazebo9'. Thus, a state vector is composed of total 65 features. We recorde the trajectories by sampling the state vectors at a 100Hz frequency.

### 1.1 DATASET

This phase aims to construct a dataset consisting of trajectories generated through random robot executions. This dataset is utilized for training the *Action Model* and the pre-training of the *Detect Model*. The robot is authorized to execute fundamental robotic actions, including movement of the mobile platform, rotation, and manipulation of joint angles in the robot arm. The robot was positioned within the environment to autonomously carry out diverse actions and engage with the

surroundings in a random manner. It is assumed that a supervisor provides action labels to the trajectories, thereby yielding a collection of labeled trajectories. To enhance diversity, we apply two heuristics while executing actions. Initially, we introduce random alterations to the spatial configurations of objects within the environment. For instance, during the execution of an action involving cube retrieval from a table, we introduce random variations to the table's position and orientation, as well as those of the cube. Subsequently, we generate diverse combinations for action execution. For instance, in the context of movement actions, the robot preforms different movement combinations.

## 2 IMPLEMENTATION DETAILS

### 2.1 TRAINING OF THE *Action Model*

Utilizing the dataset developed through random robot executions, we proceed to train the Action Model. This model comprises an LSTM layer, alongside two fully-connected blocks, each subsequently accompanied by a batch normalization layer and a ReLU activation function. To ensure regularization, a dropout layer is incorporated. The ultimate layer employs the softmax function, outputting the probabilities of prediction for each action label. Through learning from the tuple of action trajectory and label, the network discerns distinctive characteristics within various actions' trajectories, thus configuring itself as a multi-classifier primed to identify specific action types within a provided trajectory fragment.

The sequence learning window width is set to 200. Therefore, input for the *Action Model* comprises trajectory fragments, each with dimensions of $200 \times 65$. The initial layer comprises a bidirectional stacked LSTM with 128 hidden states. It is followed by two fully-connected network blocks, with hidden sizes $256 \times 128$ and $128 \times 64$ respectively. The output layer consists of a $64 \times 4$ dense layer followed by a dropout layer with a rate of $p = 0.5$. During *Action Model* training, a batch size of 256 is utilized, and the learning rate is configured to 3e-4.

### 2.2 TRAINING OF THE *Detect Model*

The *Detect Model* is trained via a meta-learning approach. The architecture of the *Detect Model* is depicted in Figure 1. Primarily, the *Detect Model* comprises a Transformer encoder, succeeded by a linear layer. The *Detect Model* encoder features key, query, and value sizes of 64, alongside a hidden size of 64. Additionally, it employs 8 heads and a dropout rate of 0.5. Subsequently, a linear layer follows, generating two logits for action failure detection.

The training process of the *Detect Model* is depicted in Figure 1. Trajectories, each associated with a robot action label, are segmented into three distinct phases: the start stage, middle stage, and end stage. Subsequently, segments from each trajectory under every action label are aggregated into a novel dataset. Randomly selecting two groups from the pool of trajectory fragments, we establish a binary training task, by labeling one group as 1 and the other as 0. A comparable data processing approach is employed during the fine-tuning phase, involving the segmentation of trajectories pertaining to specific actions labeled as successful and unsuccessful. These segmented trajectories are combined to form a compact dataset, utilized for fine-tuning that facilitates the *Detect Model* based failure detection for a specific action.

We employ the reptile meta-learning approach (Nichol et al., 2018). During pre-training, the process spans a total of 50,000 iterations, accompanied by a corresponding learning rate of 1e-3. Within each iteration, an inner iteration is performed twice, with a learning rate of 1e-2. Fine-tuning entails updating the pre-trained network through a mere 10 gradient update steps. Training for both the *Action Model* and the *Detect Model* transpires on a machine equipped with an AMD Ryzen 9 3950X processor and an RTX3090 GPU, complemented by 128 GB of RAM.

## 3 IMPLEMENTATION OF THE EXECUTION SYSTEM FOR FAILURE DETECTION

Our work involves extending the ROSPlan framework (Cashmore et al., 2015) to automate plan execution within the ROS environment, as illustrated in Figure 2. A plan comprising sequenced actions serves as input to the execution component. The *Plan Dispatching* module ensures the execution of

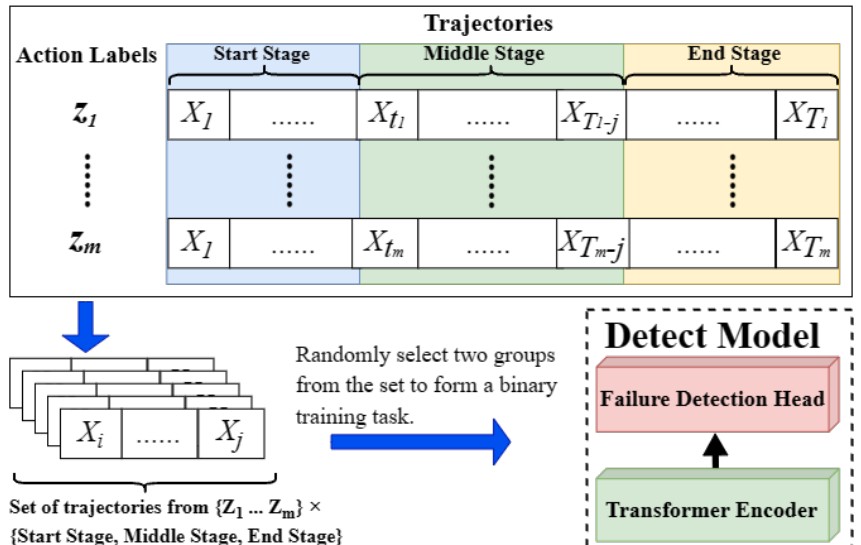

Figure 1: The training of the *Detect Model* in a meta-learning manner. The main architecture of the *Detect Model* is Transformer Encoder (lower right part).

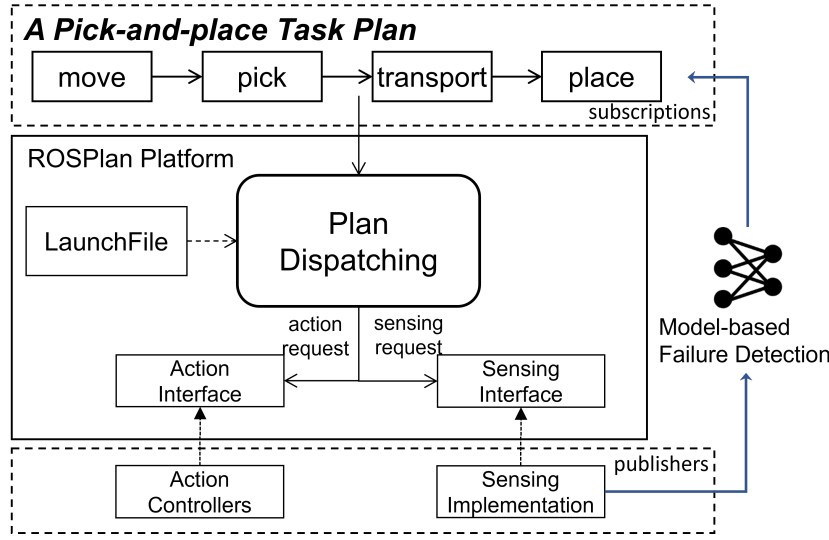

Figure 2: Our extended architecture based on ROSPlan for plan execution.

the plan. Every action within the plan is transformed into ROS messages, which supports execution in ROS environments. The interfaces, *Action Interface* and *Sensing Interface*, facilitate executing actions and determining the effect of action execution respectively.

The *Action Interface* assumes the role of executing specific robot actions, realized through the corresponding action controller. In our method, the action controller is learned by Dynamic Movement Primitives. Upon action execution completion, the *Sensing Interface* takes on the task of assessing the action's outcomes. In our method, the action failure detection is achieved by using the *Detect Model* to determine whether the execution of the action was successful or not. The relevant configurations are specified in the *LaunchFile*.

## 3.1 CODE AND VIDEO

The code and video of our method can be found here.

## 4 CASE STUDY

We offer a specific example to show how our approach acquires executable a failure-aware plan through a limited number of demonstrations.

### 4.1 SEGMENTATION INTO ACTION SEQUENCE

Upon recording the demonstration trajectories, we employ the pre-trained *Action Model* for segmentation. The segmentation results for the set of five demonstrations are depicted in Figure 3. Assuming that the user's demonstrating is relatively consistent and ideal, the sequences of the five demonstration trajectories closely resemble the task scenario.

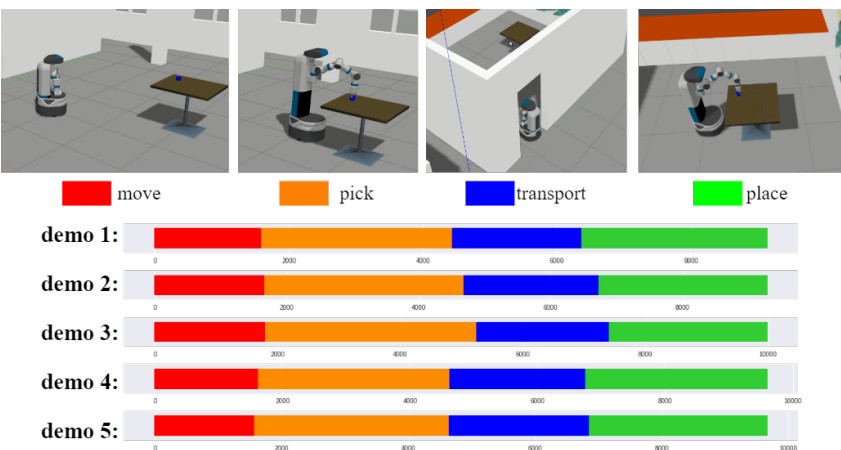

Figure 3: Action segmentation results for the 5 demonstrations. The action sequences are learned for the pick-and-place task.

After applying our sequence learning on the user demonstration trajectories, we obtain sequenced action segmentations that comprise the state observations and their corresponding action labels. $\Pi_{task}$ can be formulated as follows:

$$\langle move_1, pick_1, transport_1, place_1 \rangle$$

### 4.2 LEARNING ACTION CONTROLLERS AND FAILURE DETECTORS

We utilize Dynamic Movement Primitives (DMPs) to learn controllers for each action within $\Pi_{task}$. The DMPs are trained for each action using the five demonstration trajectories obtained from the segmentation outcomes. This procedure allows us to derive a dedicated controller for every action within $\Pi_{task}$. Subsequently, the controllers for each action are structured and integrated into a ROS program executable by the 'Fetch' robot. At the same time, we train the *Detect Model* by utilizing the data from the five demonstration trajectories. Building upon the foundation of the pre-trained *Detect Model*, the model is fine-tuned to accommodate the distinct actions. Concerning the observations associated with each action, we label the end phase of the trajectory as 1, while assigning a label of 0 to the remaining observations. Employing this modest dataset, we conduct training for the *Detect Model* through a few gradient descent iterations. After training, we can formulate an executable action sequence from $\Pi_{task}$ with the corresponding action controllers and the execution of each action is guaranteed by the fine-tuned *Detect Model*.

### 4.3 PLAN EXECUTION

We validate the generated plan by introducing a new request with the same task goal as demonstrated in a slightly altered environmental context. In this new scenario, we instruct the robot to pick up a cube from *TableA*, which is positioned differently from the demonstrated setup. The robot is then directed to transport the cube and place it onto *TableB*, located in a distinct position within the room.

We execute the plan multiple times to fulfill this new task request. Within these attempts, certain task executions are successfully accomplished, while others are prematurely terminated. The former outcomes indicate that failure detector for each action yields a positive evaluation, thus allowing for the subsequent actions to be executed. Conversely, the latter outcomes imply a negative evaluation of a failure detector for a particular action. For instance, in a given attempt, the robot might encounter problems in grasping the object due to the inherent randomness in kinematic calculations of the robotic arm. Despite a successful run of the $pick_1$ action, our deep model based detector correctly finds the anomaly by receiving a negative result from $Detector_{pick_1}$. Consequently, the execution engine is alerted to with failure and discontinues the progression of the entire task. However, we acknowledge the possibility of some actions being misjudged in terms of their execution effects. For example, actions that are correctly performed might be deemed abnormal. To comprehensively evaluate the effectiveness of our approach, larger-scale experiments are conducted in the subsequent section.

## 5 BASELINE APPROACH IMPLEMENTATION DETAILS

Current research predominantly revolves around learning task plans from demonstrations, often overlooking action failure detection. Thus, we extend the existing method proposed by Konidaris et al. (2018) for comparative purposes in action failure detection. This paper tackles the challenge of formulating abstract representations to facilitate planning in high-dimensional, continuous environments. A framework is introduced that combines high-level actions and symbolic representations to enable efficient planning in complex robotic tasks. The paper constructs a symbolic model for planning domains based on abstract subgoal options, which are high-level actions. The symbolic model construction involves identifying factors that change together, constructing a vocabulary to represent the changes, and defining operators based on the effects of the options. We implement their approach and successfully derive a task plan. The acquired plan is compatible with our execution implementation. State features correlated with the plan's action definitions, are captured by the algorithm introduced in this paper. We employ a straightforward deep model to learn the failure detector from demonstration trajectories. Since the features are derived from a specific algorithm, this method can be categorized as rule-based. Our approach mainly employs meta-learning for training, called a model-based methodology. Through experiments, it can be found that compared to the rule-based approach, our model-based approach can achieve good results. We refrain from delving into probabilistic planning. This is due to our primary focus on action failure detection, which is on the outcomes resulting from action execution, as opposed to the planning process itself.

## 6 LIMITATIONS

Implementing the approach in a real-world setting introduces additional challenges. In a simulated environment, we can readily access all environmental information through the 'Gazebo9' interface. Unlike the simulated environment, acquiring and recording real-world environmental states requires dependable environmental perception. Fortunately, current capabilities allow real-time tracking of the robot and recording of environmental information via diverse sensors. Additionally, the approach relies on dependable human-robot interaction techniques for conducting demonstrations.