# OpenReview forum: "Robot Learning from Demonstration: Enhancing Plan Execution with Failure Detection Model"
_ICLR.cc/2024/Conference — Submitted to ICLR 2024_

### Official Review · Reviewer_z6Zs · 2023-10-30

**Soundness:** 2 fair
**Presentation:** 2 fair
**Contribution:** 1 poor
**Rating:** 3
**Confidence:** 3

**Summary:**

This paper addresses concerns related to action failure during the learning-from-demonstrations process. To address this challenge, the authors design an action model to categorize actions and introduce an action-level failure detector model that employs meta-learning for failure detection. Experiments are conducted in both stationary and variable settings, potentially showcasing the efficacy of the proposed action failure models in the simulator environment.

**Strengths:**

* Detecting action failures is an interesting and crucial aspect of robot learning from demonstrations, which can enhance the stability and reliability of the learned policy.

**Weaknesses:**

* The chosen tasks are too simple to illustrate the effectiveness of the action failure model. In the meanwhile, it’s unclear if the user demonstrations are collected through teleoperation or just some rule-based demonstrations.
* There is lack of comparison and connection to the related Behaviour Cloning work (e.g. explicit imitation learning (ACT), implicit imitation learning (iBC) and some recent popular Diffusion Policy). Without a proper connection to the popular algorithms on learning from demonstration, it’s hard to evaluate the effectiveness of the proposed action failure detection model.
* For the categorization of the action, it may not be realistic for more complicated tasks.
* There is limited discussion on how to leverage the action failure model to further boost the success rate of the learned policy. Some inituitive way may be redo the action when it fails. And the evaluation metric should be the final success rate.

**Questions:**

* Is there some idea on how the approach can be used with some matured imitation learning algorithms or some other matured algorithms on learning from demonstrations?
* For the random exploration stage, is it pure-random or with some rule-based heuristic for each action type?
* For table 1, why your methods with no-ft or ft failure detector have different number on the final success? The final success should be related to the learned policy, why related to the action failure? After you detect the action failure, the process will just be terminated, right? Is this also counted as #ES?

---

### Official Review · Reviewer_kyzR · 2023-10-31

**Soundness:** 3 good
**Presentation:** 2 fair
**Contribution:** 2 fair
**Rating:** 3
**Confidence:** 3

**Summary:**

This submission focuses on robot learning by demonstration and aims to introduce a failure detection model that holistically oversees the execution of the overall plan, rather than using individual failure detection models focusing on individual actions. The proposed solution consumes a small set of human-provided demonstrations and process them in 3 stages. Initially, a sequence of actions comprising a plan is derived from the demonstrations employing a bi-directional LSTM model and a time-windowed voting method. Subsequently, action controllers are learned based on the segmented demonstrated action data, using DMPs. Upon deployment, a Detection model is employed to oversea the execution of each action by the respective controller. Interestingly, this detection model is meta-learned in view of the provided demonstrations, before being fine-tuned to each respective task.

**Strengths:**

-The submission studies the interesting problem of robot learning by demonstration, under the supervision of a failure detection model. In contrast to the plethora of "open loop" approaches in the literature, closing the loop of action execution with additional supervision for failure detection is a potentially very impactful problem setting.

-The control and supervision model are jointly trained, and the design choice of plan-level supervision can develop towards a foundational model for failure detection under a wide variety of tasks and environments, given the timely release of large scale task datasets in real-world environments: https://arxiv.org/abs/2310.08864

**Weaknesses:**

-Although the problem setup is well-formulated (Sec. 3.1), some aspects of the proposed methodology are not clearly explained in the manuscript. Most prominent example is the adoption of a meta-learning training framework in Sec. 3.4, which although appears to be one of the key contributions of the paper, is not adequately discussed (formulation of adopted loss and training pipeline details are missing).

-The experimental evaluation of the proposed approach is severely limited. Comparisons are conducted solely in simulation, with a single baseline that lacks any mechanism for failure identification. A comparison with approaches that apply failure detection to individual actions (some cited in the related work section) would be more appropriate to demonstrate the capabilities and limitations of the proposed approach.

-The manuscript claims that an extension to real-world is possible due to the high-fidelity of the Gazebo simulator. This is a very bold statement that is not backed by any experimental data.

-Some sections of the paper, (primarily the introduction and parts of the evaluation) are not well written and feature a large number of syntax and grammar errors and typos

**Questions:**

1. Please provide a more formal and detailed description of the meta-learning pipeline used to train the Detect model.

2. How does the proposed approach compare to other frameworks that adopt action-level failure detection on the tasks examined in the experimental evaluation?

3. Have the authors experimented with the application of the proposed methodology in real-world data/settings?

Presentation/Typos:

Abstract:
-cooking coffee -> making coffee
-robot randomness execution -> robot random trajectory execution (?)

Intro:
-While research (...). They (..). -> .While research (...), they (...)
-Action Model.T This (,,,)
-The fine-tuned model are able -> is
- for failure detection for the corresponding -> (...) of the corresponding
-an baseline -> a baseline

---

### Official Review · Reviewer_YtSr · 2023-10-31

**Soundness:** 2 fair
**Presentation:** 3 good
**Contribution:** 1 poor
**Rating:** 3
**Confidence:** 4

**Summary:**

This paper proposes a methodology for action-failure detection in a robotics setting. Prior work largely focuses on predicting the best action given an observation, but ignores action-failure detection which may be used to inform recovery. The authors conduct experiments in the Gazebo environment to evaluate their proposed meta-learning based framework. Experimental results show that the proposed method is able to detect action-failures better than the baselines.

**Strengths:**

- Sound motivation. The motivation behind this work is interesting and sound.
- Good presentation. The text, along with the figures, does a good job of communicating ideas and results.

**Weaknesses:**

- Limited Evaluation. The proposed methodology is only evaluated in one very simple pick-and-place task in simulation. It is very difficult to believe that the results on this one simplistic environment will transfer to a variety of tasks, let alone to the real world.
  - Furthermore, the authors claim that the "high-fidelity nature of the simulator" suggests their results will transfer to the real world. This is very difficult to believe, and completely dismisses the entirety of the sim2real literature, which attempts to address the well-known simulation-to-real gap.
- Missing baselines. The proposed methodology is not compared to any baselines from prior work.
  - Existing policies which learn from demonstrations for long-horizon tasks may implicitly learn to do failure detection -- these should be compared against as well.
  - "A solely learned action failure detector may not achieve successful generalization to the plan." Empirically demonstrating this by comparing against such a baseline would be more convincing.
- Missing ablations. A number of design choices are made in the proposed intricate method, but very limited ablations are performed to evaluate these design choices.

**Questions:**

- The plots in Figure 3 seem to be over 10 random seeds, but there don't seem to be any error bars in the plots. Why is this the case?
- The motivation behind the proposed work is that most prior work ignores failure detection, and that this can be problematic in long-horizon tasks. While the experiments design a methodology to detect failures, it is not shown how such a module can be used to improve the downstream policy. What use is a failure detection module if its result is not utilized to improve execution?

---

### Official Review · Reviewer_3kJ5 · 2023-11-01

**Soundness:** 2 fair
**Presentation:** 2 fair
**Contribution:** 2 fair
**Rating:** 3
**Confidence:** 4

**Summary:**

This papers proposes a framework that predicts action plans and detects failed execution of the plan.  The policy module is trained with imitation learning from human demonstrations, where trajectories are partitioned into segments.  The policy learns to predict discrete action labels, followed by a DMP controller to generate exact robot action.  The detection model is a transformer that predicts if the current observed states resulted from successful execution of a given action (stage).  The paper conducts experiments in a simulation environment.

**Strengths:**

The idea of prediction successful/failed action execution is indeed important.  Being able to detect failures enable the robot to replan / correct the original action plans.

**Weaknesses:**

1. Missing details:
    a. How the action are discretized into labels is unclear
    b. How to partition trajectories into action segment is unclear.
    c. No details about meta learning.
    d. How the DMP equation related to action label $z$ is unclear.
    e. Details of Detection Model are unclear.  I'm not sure if the authors train separate model per action stage, or share the same model conditioned on different action stage label.
    f. How to categorize each action into three stages is unclear.

2. I'm not sure if using other action (stages) as negative samples make sense.  Because in the practice, the failed action execution usually results in environment states different from successful execution of other actions (e.g. dropping the mug on the floor does not belong to any other successful action).  I doubt that the Detection Model works in practice.

**Questions:**

1. How are actions discretized into labels? Also, what are the representations of robot actions?

2. How to partition trajectories into action segments? Do you apply some hand-crafted heuristics, or learn a action segmentation model?

3. What's the formulation of meta learning? What are the inputs, loss functions, and optimization procedure?

4. How is the DMP equation related to action label $z$? There's no $z$ in the equation

5. Do you train separate Detection Model for each action stage, or share the same model conditioned on different action stage label?

6. How to categorize each action into grhee stages?

7. Confusing denotation
    * what does $M$ mean in the 1st paragraph of Setion 3.1: $D_M=\{X_{m, t}, z_{m, t}\}_{m=1:M, t=1:T}
    * what do $y$ and $x$ mean in the DMP equation? (Section 3.3)

---

### Meta-Review · Area_Chair_NCpN · 2023-12-05

**Metareview:**

Summary: The paper proposes a framework for detecting action failures in robot plan execution. The detection model is trained via meta-learning from a small set of segmented human demonstrations, and at deployment it predicts if the observed states resulted from successful execution of a given action. The paper conducts experiments in the simulated Gazebo environment and showcases some promising results against the chosen baselines.

Strengths:
- The motivation of action failure detection is sound, important, and well-presented.
- The paper is reasonably well written and presented clearly.

Weaknesses:
- Limited evaluation (tasks and environments): the paper only presents experiments in simulation, on a very simple pick and place task. Reviewers agree that the paper needs more compelling evaluation to demonstrate the method's ability to transfer to more varied and complex tasks or to the real world.
- Limited evaluation (baselines and ablations): the paper evaluates the method against a single baseline that lacks any mechanism for failure identification. Reviewers ask for comparison with more compelling approaches, including ones cited in related work that apply failure detection to individual actions. Moreover, a number of design choices are made that are not properly ablated.
- Some lack of clarity and missing details: multiple reviewers have unresolved questions about the proposed methodology. Additionally, some sections of the paper are rife with typos and grammatical errors.

**Justification For Why Not Higher Score:**

All reviewers agree that the evaluation task is rather toy, in a simulated environment, which does not provide evidence for the applicability and transfer of the method to the real world or to more complex tasks. The evaluation is also missing important comparisons to relevant baselines cited in the related work. Lastly, there are still many clarifications that need to be addressed. The authors did not provide a rebuttal to address any of these major weaknesses.

**Justification For Why Not Lower Score:**

N/A

---

### Decision · Program_Chairs · 2024-01-16

Reject